# Clinical indicators combined with S100A12/TLR2 signaling molecules to establish a new scoring model for coronary artery lesions in Kawasaki disease

**Yali Wu[‡], Shasha Wang[‡], Yang Zhou, Youjun Yang, Shiyu Li, Wei Yin, Yan Ding***

Department of Rheumatology and Immunology, Wuhan Children's Hospital (Wuhan Maternal and Child Healthcare Hospital), Tongji Medical College, Huazhong University of Science & Technology, Wuhan, China

‡ YW and SW are contributed equally to this work and share first authorship on this work
* dingyanmx@163.com

**Data Availability Statement:** All relevant data are within the paper and its Supporting Information files.

## Abstract

Coronary artery lesions (CALs) are the most common and serious complication of Kawasaki disease (KD), and the pathogenesis is unknown. Exploring KD-specific biomarkers and related risk factors is significant for clinical diagnosis and treatment. This study aimed to explore the feasibility of combining clinical indicators with S100A12/TLR2-associated signaling molecules for the predictive modeling of CALs in KD. A total of 346 patients (224 males and 122 females) with KD who visited the rheumatology department of Wuhan Children's Hospital between April 2022 and March 2025 were enrolled and divided into two groups according to the presence or absence of CALS (292 patients had CALs and 54 patients did not). Forty-one variables were collected from the two groups, including demographic characteristics, clinical manifestations, and laboratory data. Single nucleated cells from each patient were extracted, and the expression of the S100A12/TLR2 signal transduction-related molecules S100A12, TLR2, MYD88, and NF-κB were detected by real-time fluorescent quantitative polymerase chain reaction. Statistically significant variables were subjected to logistic regression analysis to determine the independent risk factors for KD with CALs, and a new risk score model was established to assess the predictive efficacy based on receiver operating characteristic curves. Sixteen variables significantly differed between the no-CALs and CALs groups: gender, fever duration, white blood cells (WBC), hemoglobin (HGB), Ce reactive protein (CRP), procalcitonin, serum ferritin (SF), erythrocyte sedimentation rate (ESR), fibrinogen (FIB), aspartate aminotransferase-to-alanine aminotransferase ratio (AST/ALT), serum albumin (ALB), sodium (Na), Interleukin (IL-10), tumor necrosis factor (TNF-α), S100 calcium binding protein A12 (S100A12), and Myeloid Differentiation Factor 88 (MYD88) ($p < 0.05$). After performing a univariate analysis, 12 variables (gender, fever duration, WBC, HGB, CRP, SF, ESR, FIB, AST/ALT, ALB, Na, and S100A12) were included in the multifactorial binary logistic regression, which showed that fever duration $\geq$ 6.5 days, ESR $\geq$ 46.5 mm/h, AST/ALT $\leq$ 1.51, and S100A12 $\geq$ 10.02 were independent risk factors for KD with CALs and were assigned scores of 3, 2, 1, and 2, respectively, according to the odds ratio (OR). The total score of each patient was counted, and a new

**Funding:** This study was partly financed by the Wuhan City Health and Family Planning Commission of clinical medical research major project (No.WX19M03) and Soaring Plan of Youth Talent Development in Wuhan Children's Hospital. Furthermore, it did not interfere with the study's design and collection, analysis, data interpretation, and manuscript writing. There is no additional external funding received for this study.

**Competing interests:** The authors have declared that no competing interests exist.

prediction model for KD combined with CALs was established, where < 3.5 was considered low risk and ≥ 3.5 was regarded as high risk; the sensitivity, specificity, Jorden index, and area under the curve of this scoring system were 0.667, 0.836, 0.502, and 0.838, respectively. This new scoring model has good efficacy for predicting the occurrence of KD with CALs. The expression of S100A12 was significantly increased in the CALs group and was an independent risk factor for the occurrence of CALs, and has the potential as a biomarker for predicting KD with CALs.

## Introduction

Kawasaki disease (KD) is an acute febrile disease of unknown etiology, characterized by systemic inflammation and vasculitis [1]. In contrast to Multisystem Inflammatory Syndrome in Children (MIS-C) related to Coronavirus disease 2019 (COVID-19), KD mostly occurs in children over five years of age. The most common complication of KD is coronary artery disease, especially coronary artery lesions (CALs) such as coronary artery dilation, aneurysm, coronary artery thrombosis and stenosis, life-threatening coronary aneurysm rupture, myocardial infarction, and heart failure in severe cases [2, 3]. The use of intravenous immunoglobulin (IVIG) and aspirin in KD has significantly decreased the incidence of CALs, but CALs are still observed in 5–20% of patients in the acute phase [4–6]. Nevertheless, the pathogenesis of CALS due to KD is currently unclear, and no validated biomarkers are available to predict the occurrence of this complication. Therefore, it is challenging for clinicians to predict the occurrence of CALs and implement effective intervention measures. In this study, we aimed to explore the efficacy of clinical indicators combined with S100A12/TLR2-related signaling molecules in predicting KD with CALs and provide a basis for understanding the pathogenesis of CALs, and assist clinical decision-making.

A member of the calcium-binding S100 family, S100A12 activates inflammatory responses by binding to multiple receptors extracellularly in damage-related molecular patterns [7]. Previous studies have found that neutrophils in the early stage of KD can secrete S100A12 [8] to promote KD coronary artery lesions through the synergistic activation of endothelial cells [9, 10], whereas S100A12 strictly depends on pattern recognition receptors, such as toll like receptors (TLRs), to function in vivo [11, 12]. The S100A12 and TLR signaling molecules have not yet been reported as indicators of KD. TLRs are important in innate immunity, immunodeficiency, and COVID-19 [13, 14]. The application of TLRs, cytokines, and other biomarkers in severe COVID and MIS-C (similar to KD) may provide a good example to illustrate the progress of novel biomarker application in rare or severe infections [15]. TLR2 has been reported to predict CAL progression in patients with KD [16, 17], and a study by Soo et al. found that the high expression of TLR2 in single-nucleated cells was correlated with CALs and non-response in KD [16]. Highly expressed TLR can activate the signaling mediators domain-containing adapter-induced interferon-β (TRIF) and Myeloid differentiation primary response gene 88 (MYD88), allowing nuclear factor kappa-B (NF-kB) activation to promote TNF-α, IL-1, IL-6, and other pro-inflammatory factors [18]. Previous studies have confirmed that TLR2 and MyD88 contribute to *Lactobacillus casei* extract-induced focal coronary arteritis in a mouse model of KD [19], and Seyed et al. found that *TLR2*, *TLR3*, *TLR9*, *MYD88*, and *TRIF* gene transcript levels were upregulated in KD patients before IVIG treatment and downregulated after treatment [20]. In addition, several studies have found that NF-κB is involved in the development and progression of KD by participating in the inflammatory response, regulating

the release of inflammatory factors [21], participating in immune activation [22], and inducing vascular endothelial damage [23]. Therefore, we envision that S100A12/TLR2 may activate the immune response of KD and may be involved in the development of CALs in KD by inducing the high expression of NF-κB by MYD88.

## Materials and methods

### Study population

Cohort study. Children with KD admitted to the Department of Rheumatology and Immunology of Wuhan Children's Hospital in Hubei province, China, from April 1, 2022, to March 31, 2023, were used as the cohort population. They were effectively treated and followed up for eight weeks after discharge and were divided into no-CALs and CALs groups according to the occurrence of CALs, all with a complete medical history and laboratory and cardiac ultrasound data.

KD inclusion criteria: 1. meeting the KD diagnostic criteria published by the American Heart Association in 2017 [24]; 2. case information; 3. age < 18 years; 4. patients with a first diagnosis of KD and not receiving IVIG treatment.

The KD exclusion criteria were as follows: 1. incomplete treatment data; 2. underlying coronary artery disease; 3. previous use of hormonal drugs within four weeks; 4. secondary infection; 5. lost to follow-up; 6. immunodeficiency.

CALs were defined according to the Japanese Ministry of Health criteria [25, 26] using echocardiography performed for each KD patient: an internal lumen diameter greater than 3.0 mm in children < 5 years of age or greater than 4.0 mm in children ≥ 5 years of age or an internal diameter at least 1.5 times larger than the diameter of the adjacent segment, if the morphology of the coronary lumen was obviously irregular, or if the z-score ≥ 2.5.

This study was approved by the Institutional Review Board of Wuhan Children's Hospital, Tongji Medical College, Huazhong University of Science and Technology (NO. 2022R053-E01), and all guardians of the children signed an informed consent form.

Patients with KD admitted to the hospital from April 1, 2022, to March 31, 2023, were recruited into the study cohort according to the inclusion and exclusion criteria and after obtaining informed consent. Cardiac ultrasonography was performed to assess their coronary arteries from weeks 0, 1, 4, and 8 after the diagnosis of KD, and patients with CALs were found to require more frequent monitoring than the four times mentioned above. Missed visits were excluded, and 292 and 54 patients in the non-CAL groups and CAL, respectively, were included (Fig 1).

### Data collection

The data platform and medical record system of Wuhan Children's Hospital acquired the information of 346 patients and collected baseline information (gender, age, and body mass index [BMI]), clinical manifestations (fever duration, rash, conjunctivitis, lip and tongue changes, lymphadenopathy, and extremity changes), laboratory results (white blood cells [WBC], hemoglobin [Hb], platelets [PLT], neutrophil-to-lymphocyte ratio [NLR = neutrophils (×109/L)/lymphocytes (×109/L)], C reactive protein [CRP], procalcitonin [PCT], erythrocyte sedimentation rate [ESR], serum ferritin [SF], fibrinogen [Fib], pusuria, aspartate aminotransferase-to-alanine aminotransferase ratio [AST/ALT], lactate dehydrogenase [LDH], creatine kinase-MB [CK-MB], serum albumin [ALB], total bilirubin [TBIL], direct bilirubin [DBIL], serum creatinine [Scr], urea nitrogen [BUN], serum Na, CD3+CD4 +T, CD3+CD8+T, CD19+B, CD4/CD8+T, IgM, IgG, IgA, IL-2, IL-4, IL-6, IL-10, IFNγ, and

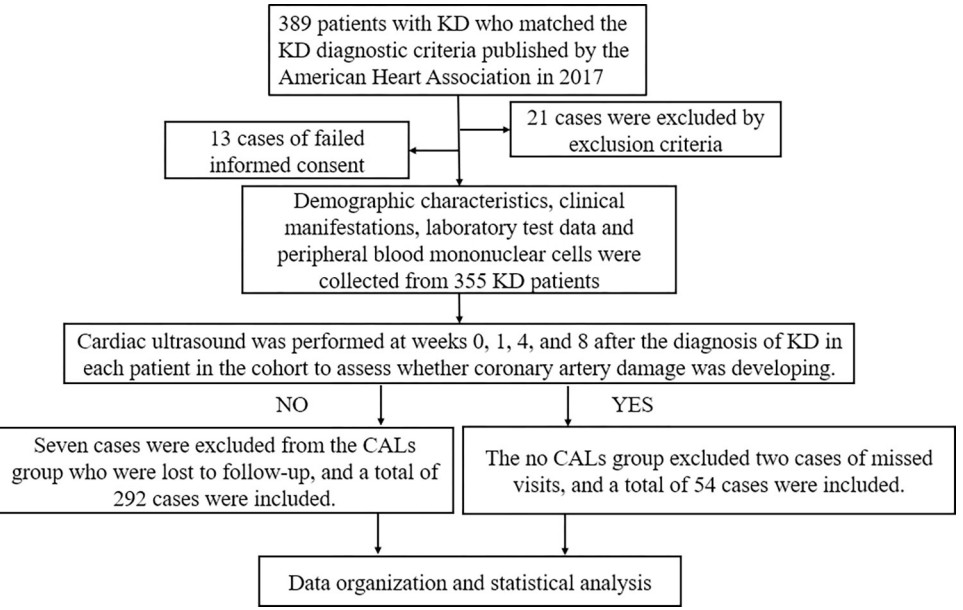

**Fig 1. Flow chart of study subject inclusion.** KD, Kawasaki disease; CALs, Coronary artery lesions.

TNFα), cardiac ultrasound (coronary artery diameter and Z-value), which were entered into EXCEL tables.

To obtain peripheral blood mononuclear cells (PBMCs), peripheral venous blood was taken from all KD patients in the cohort, anticoagulated using heparin, and added to the stratification solution. PBMCs were extracted after centrifugation and washed twice with sterile phosphate-buffered saline, and the cells were resuspended in RPMI-1640 culture medium and stored at 4˚C.

RNA was extracted from the PBMC samples using TRIzol reagent and stored at -70˚C, and the RNA concentration and purity were measured with the NanoDrop ND-1000 (Thermo Fisher Scientific Inc., Massachusetts, USA) spectrophotometer. RNAs were converted into cDNA by reverse transcription using the PrimeScript TM RT Reagent kit (Vazyme Biotech [Nanjing] Co., Ltd. CHINA). Following cDNA synthesis, real-time polymerase chain reaction (PCR) and relative quantification were performed using Premix Ex Taq SYBR (Vazyme Biotech [Nanjing] Co., Ltd.) to evaluate S100A12, TLR2, MYD88, and NF-κB gene expression. A SYBR Green Master Mix (20 μl; Invitrogen; Vazyme Biotech [Nanjing] Co., Ltd.) was used for real-time PCR amplification. The amplification conditions were as follows: 95˚C for 30 s, followed by 40 cycles of 95˚C for 3–10 s and 60˚C for 10–30 s. The expression levels were calculated using the $2^{-\Delta\Delta CT}$ method, and the levels were normalized against glyceraldehyde-3-phosphate dehydrogenase (GAPDH) as the control blank gene. Results are expressed as fold-changes. Gene-specific primer sequences were designed by Sangon Biotech Co. Ltd. (Shanghai, China) and are listed in Table 1. All experiments were performed at least three times.

## Statistical analysis

Data analysis was performed using SPSS 22. The measurement data were tested for normality, and non-normally distributed measurement data were expressed as M (Q1, Q3), with comparisons between groups made by Mann-Whitney U tests. Count data were expressed as the

**Table 1. The sequence of primers.**

| Primer | Sequence |
|---|---|
| S100A12 Forward | AGCATCTGGAGGGAATTGTCA |
| S100A12 Reverse | GCAATGGCTACCAGGGATATGAA |
| TLR2 Forward | ATCCTCCAATCAGGCTTCTCT |
| TLR2 Reverse | GGACAGGTCAAGGCTTTTTACA |
| MYD88 Forward | GGCTGCTCTCAACATGCGA |
| MYD88 Reverse | CTGTGTCCGCACGTTCAAGA |
| NF-κB Forward | AACAGAGAGGATTTCGTTTCCG |
| NF-κB Reverse | TTTGACCTGAGGGTAAGACTTCT |
| GAPDH Forward | GCACCGTCAAGGCTGAGAAC |
| GAPDH Reverse | TGGTGAAGACGCCAGTGGA |

number of cases and compared between groups using the four-compartment table χ2 test, and all differences were considered statistically significant at $p < 0.05$. Differential variables were subjected to a one-way logistic analysis, and $p < 0.05$ was included in a multifactorial logistic model to screen valuable risk factors and score them according to OR values to establish a predictive scoring model. The total score was calculated for each group of patients, and the maximum Youden index corresponding to the total score cutoff value, sensitivity, and specificity were derived using the subject receiver operating characteristic (ROC) curve. Statistical significance was set at $p < 0.05$.

## Results

### Baseline information and clinical presentation in the no-CALs and CALs groups

Among the 346 patients included in the study, 224 were male and 122 were female, with the no-CALs and CALs groups significantly differing in their sex ratios (180/112 versus 44/10 males/females for no-CALs and CAL, respectively; $p = 0.005$) and fever duration (5 [5, 6], 6.5 [5, 8]; $p < 0.001$). There were no statistically significant differences between the two groups in age, BMI, rash, conjunctival congestion, lip and tongue changes, neck lymph node enlargement, and terminal changes in the limbs (Table 2).

**Table 2. Baseline and clinical presentation of the No CALs and CALs groups.**

| Items | Total | No CALs | CALs | U/χ2 | P |
|---|---|---|---|---|---|
| No. of patients | 346 | 292 | 54 | | |
| Gender (M/F) | 224/122 | 180/112 | 44/10 | 7.856 | 0.005 |
| Age(years) | 3.41[1.6,5.58] | 3.53[1.74,5.72] | 3.1[1.05,4.57] | -1.666 | 0.096 |
| BMI | 15.47[14.13,16.84] | 15.51[14.14,16.8] | 15.19[14.04,17.57] | -.130 | 0.897 |
| Fever duration(days) | 6[5,7] | 5[5,6] | 6.5[5,8] | -4.267 | 0.000 |
| Rash (n) | 279 | 233 | 46 | 0.848 | 0.357 |
| Conjunctivitis (n) | 307 | 259 | 48 | 0.002 | 0.968 |
| Lip and tongue changes(n) | 298 | 247 | 51 | 3.705 | 0.054 |
| Lymphadenopathy (n) | 233 | 201 | 32 | 1.9 | 0.168 |
| Extremity changes(n) | 248 | 211 | 37 | 0.314 | 0.575 |

BMI, Body Mass Index.

## Laboratory data for the no-CALs and CALs groups

A total of 32 laboratory tests were analyzed, and significant differences ($p < 0.05$) were found between the no-CALs and CALs groups for WBC, Hb, CRP, PCT, ESR, SF, FIB, AST/ALT, ALB, Na, IL-10, and TNF-α. There were no significant differences between the two groups for PLT, NLR, pusuria, LDH, CK-MB, TBIL, DBIL, Scr, BUN, CD3+CD4+T counts, CD3+CD8+T counts, CD19+B counts, CD4/CD8+T, IgM, IgG, IgA, IL-2, IL-4, IL-6, and IFNγ ($p > 0.05$; Table 3).

**Table 3. Laboratory test results between no CALs and CALs groups.**

| Variable | No CALs | CALs | U/t | P |
|---|---|---|---|---|
| WBC (10$^9$/L) | 10.47[7.56,14.58] | 12.36[9.39,16.5] | -2.430 | .015 |
| Hb (g/L) | 109[103,116] | 102[94.75,111] | -3.560 | .000 |
| PLT count (10$^9$/L) | 324.5[243.25,424.75] | 363.5[249,475.75] | -1.776 | .076 |
| NLR | 2.69[1.32,5.41] | 2.67[1.32,6.15] | -.233 | .816 |
| CRP (mg/L) | 47.5[29.03,88.92] | 83[40.4,121] | -2.854 | .004 |
| PCT (ng/ml) | 0.44[0.17,1.42] | 0.76[0.25,3.06] | -2.308 | .021 |
| ESR (mm/h) | 32.5[14,59] | 68[36.75,88] | -4.675 | .000 |
| SF (ng/ml) | 143.79[104.1,207.44] | 187.95[146.74,275.69] | -3.471 | .001 |
| FIB (g/L) | 4.78[3.74,6.11] | 6.02[4.79,7.02] | -3.897 | .000 |
| Pusuria (n) | 30 | 8 | .961 | .327 |
| AST/ALT | 1.8[1.08,2.5] | 0.86[0.41,1.53] | -5.137 | .000 |
| LDH (U/L) | 316[262,408] | 288.5[243.5,400.5] | -1.126 | .260 |
| CKMB(U/L) | 27[20,39] | 24[17,32] | -1.916 | .055 |
| ALB (g/L) | 39.1[35.83,41.4] | 35.9[31.43,39.9] | -3.880 | .000 |
| TBIL (μmol/L) | 6.25[4.3,9.28] | 6.35[4.2,11.93] | -.570 | .569 |
| DBIL (μmol/L) | 2.7[1.9,3.7] | 2.65[2.05,7.03] | -1.335 | .182 |
| Scr (μmol/L) | 27.95[23.23,34.88] | 27.85[23.28,31.45] | -.773 | .439 |
| BUN (μmol/L) | 3.3[2.6,4] | 3.4[2.65,4.2] | -.386 | .700 |
| Na (mmol/L) | 137.9[135.63,139.68] | 136.6[134.38,138.23] | -2.966 | .003 |
| CD3+CD4+T (cells/μl) | 327[105.5,890.75] | 453[164,1040.25] | -1.107 | .268 |
| CD3+CD8+T (cells/μl) | 543.5[321.25,818.75] | 499[253.25,815] | -1.131 | 0.258 |
| CD19+B (cells/μl) | 752[486.75,1228] | 785[580.25,1355] | -1.14 | 0.254 |
| CD4/CD8+T | 1.7[1.18,2.34] | 2.05[1.4,2.42] | -1.799 | .072 |
| IgM (g/L) | 0.97[0.73,1.27] | 0.9[0.69,1.38] | -.675 | .499 |
| IgG (g/L) | 7.24[5.61,9.04] | 7.29[5.08,9.93] | -.118 | .906 |
| IgA (g/L) | 0.82[0.46,1.27] | 0.81[0.49,1.52] | -.377 | .706 |
| IL2 (pg/mL) | 3.67[2.95,4.88] | 3.49[2.17,4.63] | -1.440 | .150 |
| IL4 (pg/mL) | 3.42[2.7,4.44] | 3.03[2.21,4.1] | -1.565 | .117 |
| IL6 (pg/mL) | 74.25[31.02,146.93] | 108.03[24.75,350.88] | -1.443 | .149 |
| IL10 (pg/mL) | 12.41[6.54,21.92] | 16.79[8.49,33.76] | -2.301 | .021 |
| IFNγ (pg/mL) | 4.2[2.59,8.23] | 4.3[2.76,8.62] | -.433 | .665 |
| TNFα (pg/mL) | 5.39[3.52,9.52] | 4.56[2.68,6.5] | -2.234 | .025 |

WBC, white blood cell; Hb, Haemoglobin; PLT, platelet; NLR, neutrophil-to-lymphocyte ratio = neutrophils (×10$^9$/L)/lymphocytes (×10$^9$/L); CRP, C reactive protein; PCT, procalcitonin; ESR, erythrocyte sedimentation rate; SF, serum ferritin; FIB, Fibrinogen. ALT, alanine aminotransferase; AST, aspartate transaminase; LDH, lactate dehydrogenase; CK-MB, creatine kinase-MB; ALB, serum albumin; TBIL, total bilirubin; DBIL, direct bilirubin; *Scr, Serum creatinine, BUN, Urea nitrogen.*

**Table 4. Expression of signaling molecules between no CALs and CALs groups.**

| Variable | No CALs | CALs | U | P |
|---|---|---|---|---|
| S100A12 | 5.08[2.9,7.95] | 10.52[3.72,17.28] | -4.450 | .000 |
| TLR2 | 1.67[1.25,2.27] | 1.59[0.88,2.3] | -.869 | .385 |
| MYD88 | 2.35[1.27,3.12] | 2.86[1.35,5.23] | -2.069 | .039 |
| NFKB | 1.18[0.71,2.09] | 1.27[0.64,2.71] | -.548 | .584 |

### Expression of S100A12, TLR2, MYD88, and NF-κB in the no-CALs and CALs groups

The expression of A100A12 and MYD88 in the CALs group was significantly higher than that in the no-CALs group ($p < 0.05$); however, differences in the expression of NF-κB and TLR2 between the two groups were not statistically significant (Table 4).

### Independent risk factors for the development of CALs in KD

A one-way logistic regression analysis was performed for each of the 16 variables with statistically significant differences between groups, in which PCT, IL-10, TNF-α, and MYD88 were not statistically significant in KD with CALs ($p > 0.05$). Twelve independent variables (sex, fever duration, WBC, Hb, CRP, Ferr, ESR, Fib, AST/ALT, ALB, Na, and S100A12) were subjected to a multivariate binary logistic regression analysis, which determined that fever duration, ESR, AST/ALT, and S100A12 were independent predictors of KD CALs (Table 5).

### Scoring model development

The independent risk factors fever duration, ESR, AST/ALT, and S100A12 were obtained as cut-off values according to ROC analysis and converted into dichotomous variables; then, a scoring system was established according to the OR values: fever duration $\geq 6.5$ d (3 points), ESR $\geq 46.5$ (2 points), AST/ALT $\leq 1.52$ (1 point), and S100A12 $\leq 10.02$ (2 points). The total

**Table 5. Logistic regression analysis to identify independent factors predicting KD with CALs.**

| Variable | Univariate | | Multivariate | |
|---|---|---|---|---|
| | Curde OR (95%CI) | P | adjusted OR (95%CI) | P |
| Gender | 2.738[1.325,5.658] | 0.007 | | |
| Fever duration (days) | 1.455[1.218,1.739] | .000 | 1.432[1.117,1.836] | .005 |
| White blood cell (10⁹/L) | 1.06[1.014,1.109] | .011 | | |
| Hb (g/L) | 0.966[0.943,0.99] | .006 | | |
| CRP (mg/L) | 1.008[1.003,1.014] | .001 | | |
| PCT (ng/ml) | 1.038[0.97,1.111] | .282 | | |
| Ferr (ng/ml) | 1.001[1,1.001] | .072 | | |
| ESR (mm/h) | 1.024[1.014,1.035] | .000 | 1.016[1.002,1.03] | .021 |
| FIB (g/L) | 1.433[1.189,1.727] | .000 | | |
| AST/ALT | 0.417[0.287,0.606] | .000 | 0.462[0.288,0.742] | .001 |
| ALB (g/L) | 0.892[0.842,0.945] | .000 | | |
| Na (mmol/L) | 0.889[0.817,0.966] | .006 | | |
| IL-10 (pg/mL) | 1.004[0.999,1.009] | .142 | | |
| TNF-α (pg/mL) | 0.989[0.963,1.016] | .413 | | |
| S100A12 | 1.112[1.069,1.156] | .000 | 1.112[1.063,1.163] | .000 |
| MYD88 | 1.054[0.984,1.128] | .134 | | |

**Table 6. Cut-off values and points after continuous variables are converted to dichotomous variables.**

| Variable | Cut-off point | Sensitivity | Specificity | OR (95%CI) | p | Point |
|---|---|---|---|---|---|---|
| Fever duration (days) | 6.5000 | .500 | 0.788 | 1.432[1.117,1.836] | .005 | 3 ($>$ = 6.5), 0($<$0.6.5) |
| ESR (mm/h) | 46.5000 | .704 | 0.647 | 1.016[1.002,1.03] | .021 | 2 ($>$ = 46.5), 0($<$46.5) |
| AST/ALT | 1.5147 | .759 | 0.630 | 0.462[0.288,0.742] | .001 | 1($<$ = 1.5147),0($>$1.5147) |
| S100A12 | 10.0155 | .556 | 0.880 | 1.112[1.063,1.163] | .000 | 2 ($>$ = 10.0155), 0($<$10.0155) |

score was calculated for each individual patient, and the ROC analysis was performed to obtain the cut-off values, where $<$ 3.5 was considered a low risk for CAL and $\geq$ 3.5 was considered a high risk for CALs. The following were obtained for this scoring system: sensitivity: 0.667, specificity: 0.836, Jorden index: 0.502, positive predictive value: 66.67%, negative predictive value: 83.56%, and area under the curve (AUC): 0.838, in addition to the ROC curve (Table 6 and Fig 2).

## Discussion

It has been more than 50 years since the initial diagnostic criteria for KD were established [27], and although many studies have been conducted, the potential for fatal complications still exists for patients with KD combined with CALS, which has become the most common childhood-acquired cardiovascular disease in developed countries and regions [24]. In recent years, developments in molecular biology and clinical medicine techniques have enabled the discovery of biomarkers as predictors of KD with CALs [28].

In our study, we explored the combined clinical indicators S100A12, TLR2, MYD88, and NF-κB to predict the occurrence of KD combined with CALs, and we found S100A12, fever duration, ESR, and AST/ALT were independent risk factors. Previously, Helmut et al. [29] found that the inverse regulation of both sRAGE and its proinflammatory ligand S100A12 seemed to be a relevant molecular mechanism promoting systemic inflammation. Srivastava et al. [30], in their study on an integrated in-silico approach for exploring potential biomarker genes and pathways in KD, identified S100A12 as one of the pivotal genes with high connectivity. Armaroli et al. [9] identified that S100A12 was a highly expressed mediator in KD sterile inflammation. The above studies provide a theoretical basis for the involvement of S100A12 in the occurrence of CALs, while our present study provides real-world evidence; thus, we expect S100A12 to be a useful biomarker of CALs.

The longer the duration of fever in patients with KD, the higher the risk of prolonged inflammatory stimulation and, consequently, severe vasculitis and aneurysms, which were confirmed as risk factors for coronary artery involvement in several studies [31]. Wang et al. [32] found that fevers lasting $\geq$ 8 days were a predictor of CALs in KD, and Kim et al. [33] confirmed that fever duration $\geq$ 7 days was a predictor of CALs in patients with KD, while we concluded that fever duration $\geq$ 6.5 days was an independent risk factor. Our result was slightly shorter than the fever duration length determined in previous studies, which may be related to the current improvement in the ability of pediatricians in China to recognize KD and initiate treatment early once it is diagnosed at this stage [34].

ESR is a recognized marker of the inflammatory response and is susceptible to the size, shape, and number of red blood cells and immunoglobulin levels, making it more meaningful to obtain values prior to gamma globulin treatment. Previous studies have shown that elevated ESR can, to some extent, reflect the inflammatory response in the coronary arteries, and Junyan et al. [35] and Tian et al. [35] confirmed that elevated ESR was a predictor of KD complicated by CALs. The latter authors proposed a sensitivity of 53.26% and specificity of 64.14%

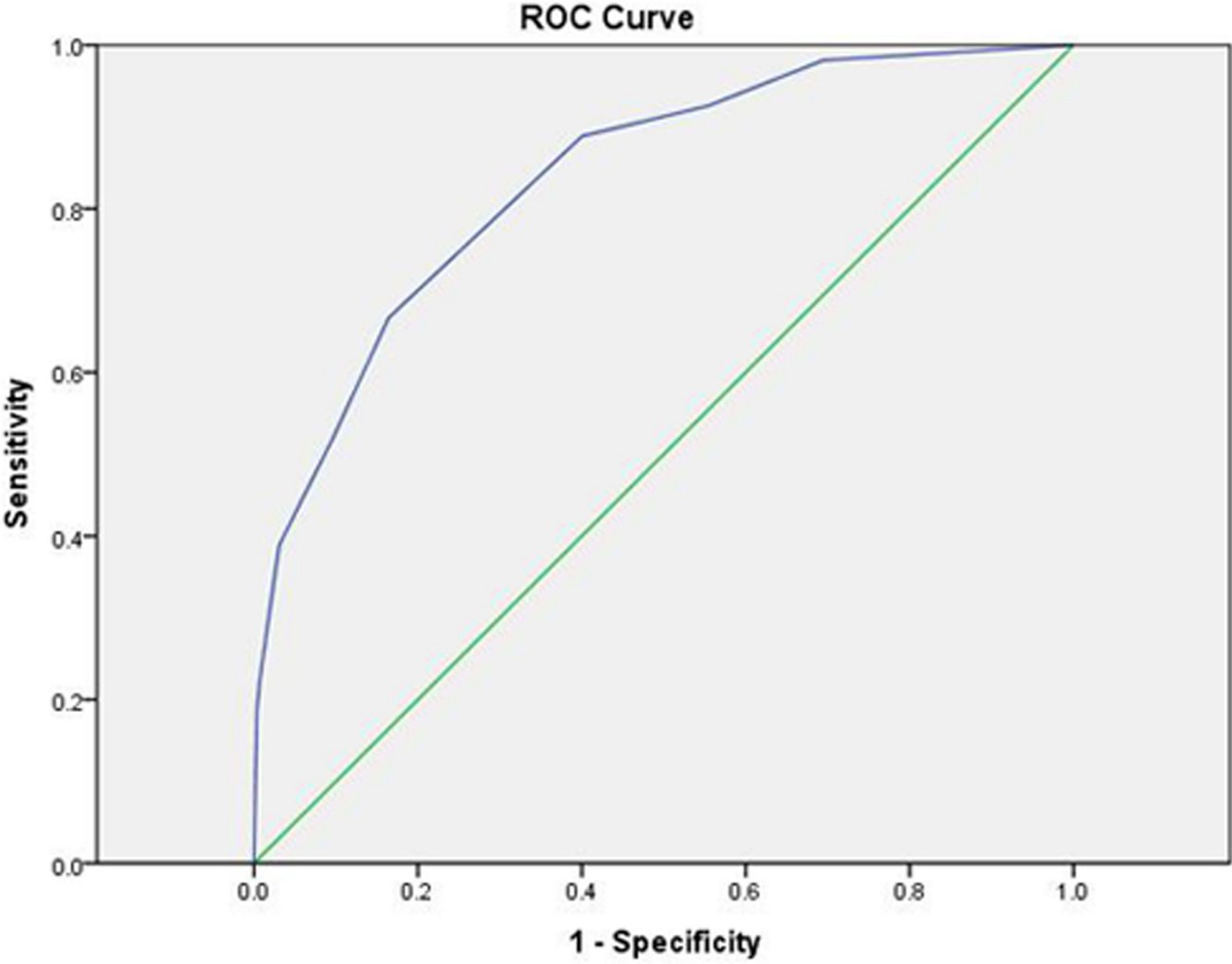

**Fig 2. ROC curves for prediction of KD with CALs by the new scoring model, AUC: 0.838(95% CI, 0.781–0.895).**

for the prediction of CALs using ESR at a critical value > 75 mm/h. In our study, ESR at a critical value ≥ 46.5 mm/h predicted a sensitivity of 70.4% and specificity of 64.7%, which is a higher sensitivity than found in that previous study.

AST is widely distributed in the mitochondria of muscle, brain, lung, kidney, and liver cells, whereas ALT is mainly distributed in the cytoplasm of hepatocytes [36]. The AST/ALT ratio is often used to assess liver function and reflect the severity of liver disease, and in their KD study, Wang et al. [37] confirmed that the smaller the AST/ALT ratio, the stronger the acute inflammatory response and the greater the risk of coronary artery injury and IVIG resistance; the ratio may therefore be a good predictor of coronary artery injury in patients with KD by reflecting the intensity of inflammation. Cao et al. [38] found that the AST/ALT ratio is a risk factor for CALs but is not associated with CAL progression. The current study is the third to confirm that AST/ALT is an independent risk factor for concomitant CALs in patients with KD. Further studies are needed to focus on coronary artery changes in patients with CALs after eight weeks.

In this study, a new scoring model for CALs in KD was established by combining clinical indicators with S100A12/TLR2 signaling molecules. Fever duration $\geq$ 6.5 days (3 points), ESR $\geq$ 46.5 mm/h (2 points), AST/ALT $\leq$ 1.51 (1 point), and S100A12 $\geq$ 10.02 (2 points) were independent risk factors for KD CALs, with scores < 3.5 presenting as low risk and $\geq$ 3.5 presenting as high risk. This scoring system had a sensitivity of 0.667, a specificity of 0.836, a Jorden index of 0.502, an AUC of 0.838, and good predictive efficacy. However, there are limitations in this study, one of which is the single method of signal molecule detection, as protein blotting and a flow cytometry assay were not performed for the quantitative validation of the protein. Additionally, the cohort study may have had a bias in the sample selection. Another limitation was the different time intervals for obtaining samples, and the sedation of gamma globulin may have introduced bias in the results. Finally, the sample size of the single-center exploratory study was small, so the results of the research were easily influenced by the objective conditions [39] and the population composition of the laboratory. In future research, it is necessary to expand the sample size for model validation and extrapolate the results across multiple centers.

## Supporting information

**S1 Data.**
(XLSX)

**S2 Data.**
(XLSX)

## Acknowledgments

### Ethic approval and consent to participate

The study was approved by the institutional review board of Wuhan Children's Hospital, Tongji Medical College, Huazhong University of Science & Technology (NO.2022R053-E01). Written informed consent was obtained from the legal guardians of the patient.

## Author Contributions

**Conceptualization:** Shasha Wang, Yan Ding.

**Data curation:** Yali Wu, Shasha Wang, Youjun Yang.

**Formal analysis:** Yali Wu.

**Methodology:** Yali Wu, Yang Zhou, Shiyu Li.

**Project administration:** Wei Yin.

**Software:** Yali Wu, Youjun Yang, Shiyu Li.

**Visualization:** Wei Yin.

**Writing – original draft:** Yali Wu, Youjun Yang.

**Writing – review & editing:** Yan Ding.

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
