## [Decision Letter · Decision Letter 0]

4 Sep 2023

PONE-D-23-26094Clinical indicators combined with S100A12/TLR2 signaling molecules to establish a new scoring model for coronary artery lesions in Kawasaki diseasePLOS ONE

Dear Dr. Ding,

Thank you for submitting your manuscript to PLOS ONE. After careful consideration, we feel that it has merit but does not fully meet PLOS ONE’s publication criteria as it currently stands. Therefore, we invite you to submit a revised version of the manuscript that addresses the points raised during the review process.

We look forward to receiving your revised manuscript.

Kind regards,

Benjamin M. Liu, MBBS, PhD, D(ABMM), MB(ASCP)

Academic Editor

PLOS ONE

Journal Requirements:

● A clean copy of the edited manuscript (uploaded as the new *manuscript* file)"

3. Thank you for stating in your Funding Statement: "This work was supported partly by the Wuhan City Health and Family Planning Commission of clinical medical research major project (No.WX19M03) and Soaring Plan of Youth Talent Development in Wuhan Children's Hospital".

4. Thank you for stating the following financial disclosure: "YES

This work was supported partly by the Wuhan City Health and Family Planning Commission of clinical medical research major project (No.WX19M03) and Soaring Plan of Youth Talent Development in Wuhan Children's Hospital".

Please state what role the funders took in the study.  If the funders had no role, please state: ""The funders had no role in study design, data collection and analysis, decision to publish, or preparation of the manuscript.

Reviewers' comments:

Reviewer's Responses to Questions

**Comments to the Author**

1. Is the manuscript technically sound, and do the data support the conclusions?

Reviewer #1: Yes

2. Has the statistical analysis been performed appropriately and rigorously? 

Reviewer #1: Yes

3. Have the authors made all data underlying the findings in their manuscript fully available?

Reviewer #1: Yes

4. Is the manuscript presented in an intelligible fashion and written in standard English?

Reviewer #1: Yes

5. Review Comments to the Author

Reviewer #1: 1. Introduction section: there is only one paragraphs in this part, which should be improved. At least two more paragraphs should be added to discuss the following points: S100A12 and TLR signaling molecules have not yet reported as an indicators of Kawasaki diseases. However, TLRs are know to play important roles in innate immunity and involvement with immunodeficiency and COVID-19. The application of TLRs and cytokine and other biomarkers in severe COVID and MIS-C (similar to Kawasaki) may set a good example to illustrate the progress of novel biomarker application in rare or severe infections, which will allow the readers to better understand the novelty of the authors' work on Kawasaki disease. The authors should introduce/discuss the above-mentioned and the following key points based on the following references: role of TLRs in human diseases (e.g. different infectious diseases, such as HCV or COVID), host immune/inflammatory biomarkers (TLRs, cytokines), pros and cons of biomarker detection, and differentiation between Kawasaki and MIS-C.

Wei D, Li NL, Zeng Y, Liu B, Kumthip K, Wang TT, Huo D, Ingels JF, Lu L, Shang J, Li K. The Molecular Chaperone GRP78 Contributes to Toll-like Receptor 3-mediated Innate Immune Response to Hepatitis C Virus in Hepatocytes. J Biol Chem. 2016 Jun 3;291(23):12294-309. doi: 10.1074/jbc.M115.711598. Epub 2016 Apr 20. PMID: 27129228; PMCID: PMC4933277.

Liu BM, Martins TB, Peterson LK, Hill HR. Clinical significance of measuring serum cytokine levels as inflammatory biomarkers in adult and pediatric COVID-19 cases: A review. Cytokine. 2021 Jun;142:155478. doi: 10.1016/j.cyto.2021.155478. Epub 2021 Feb 23. PMID: 33667962; PMCID: PMC7901304.

McCrindle BW, Rowley AH, Newburger JW, Burns JC, Bolger AF, Gewitz M, Baker AL, Jackson MA, Takahashi M, Shah PB, Kobayashi T, Wu MH, Saji TT, Pahl E; American Heart Association Rheumatic Fever, Endocarditis, and Kawasaki Disease Committee of the Council on Cardiovascular Disease in the Young; Council on Cardiovascular and Stroke Nursing; Council on Cardiovascular Surgery and Anesthesia; and Council on Epidemiology and Prevention. Diagnosis, Treatment, and Long-Term Management of Kawasaki Disease: A Scientific Statement for Health Professionals From the American Heart Association. Circulation. 2017 Apr 25;135(17):e927-e999. doi: 10.1161/CIR.0000000000000484. Epub 2017 Mar 29. Erratum in: Circulation. 2019 Jul 30;140(5):e181-e184. PMID: 28356445.

Liu B, Hill HR. Role of Host Immune and Inflammatory Responses in COVID-19 Cases with Underlying Primary Immunodeficiency: A Review. J Interferon Cytokine Res. 2020 Dec;40(12):549-554. doi: 10.1089/jir.2020.0210. PMID: 33337932; PMCID: PMC7757688.

2. For inclusion criteria, what did the author mean by "complete data"?

3. Is the statistical analysis two-tailed or one-tailed?

4. What reverse transcriptase (RT) did the authors use? What internal control gene did the RT-PCR of S100A12, TLR2, MYD88, and NFKB benchmark to? The differences in the expression of NF-κB and TLR2 between the two groups were not statistically significant. What are the limit of detection or analytical sensitivity of these RT-PCR? According to the following ref, different RTs with varied efficiency may cause unreliable results. Please discuss this based on this reference.

Liu B, Forman M, Valsamakis A. Optimization and evaluation of a novel real-time RT-PCR test for detection of parechovirus in cerebrospinal fluid. J Virol Methods. 2019 Oct;272:113690. doi: 10.1016/j.jviromet.2019.113690. Epub 2019 Jul 5. PMID: 31283959.

6. PLOS authors have the option to publish the peer review history of their article (what does this mean?). If published, this will include your full peer review and any attached files.

Reviewer #1: No

---

## [Author Response · Author response to Decision Letter 0]

25 Sep 2023

Reviewer #1: 1. Introduction section: there is only one paragraphs in this part, which should be improved. At least two more paragraphs should be added to discuss the following points: S100A12 and TLR signaling molecules have not yet reported as an indicators of Kawasaki diseases. However, TLRs are know to play important roles in innate immunity and involvement with immunodeficiency and COVID-19. The application of TLRs and cytokine and other biomarkers in severe COVID and MIS-C (similar to Kawasaki) may set a good example to illustrate the progress of novel biomarker application in rare or severe infections, which will allow the readers to better understand the novelty of the authors' work on Kawasaki disease. The authors should introduce/discuss the above-mentioned and the following key points based on the following references: role of TLRs in human diseases (e.g. different infectious diseases, such as HCV or COVID), host immune/inflammatory biomarkers (TLRs, cytokines), pros and cons of biomarker detection, and differentiation between Kawasaki and MIS-C.

Wei D, Li NL, Zeng Y, Liu B, Kumthip K, Wang TT, Huo D, Ingels JF, Lu L, Shang J, Li K. The Molecular Chaperone GRP78 Contributes to Toll-like Receptor 3-mediated Innate Immune Response to Hepatitis C Virus in Hepatocytes. J Biol Chem. 2016 Jun 3;291(23):12294-309. doi: 10.1074/jbc.M115.711598. Epub 2016 Apr 20. PMID: 27129228; PMCID: PMC4933277.

Liu BM, Martins TB, Peterson LK, Hill HR. Clinical significance of measuring serum cytokine levels as inflammatory biomarkers in adult and pediatric COVID-19 cases: A review. Cytokine. 2021 Jun;142:155478. doi: 10.1016/j.cyto.2021.155478. Epub 2021 Feb 23. PMID: 33667962; PMCID: PMC7901304.

McCrindle BW, Rowley AH, Newburger JW, Burns JC, Bolger AF, Gewitz M, Baker AL, Jackson MA, Takahashi M, Shah PB, Kobayashi T, Wu MH, Saji TT, Pahl E; American Heart Association Rheumatic Fever, Endocarditis, and Kawasaki Disease Committee of the Council on Cardiovascular Disease in the Young; Council on Cardiovascular and Stroke Nursing; Council on Cardiovascular Surgery and Anesthesia; and Council on Epidemiology and Prevention. Diagnosis, Treatment, and Long-Term Management of Kawasaki Disease: A Scientific Statement for Health Professionals From the American Heart Association. Circulation. 2017 Apr 25;135(17):e927-e999. doi: 10.1161/CIR.0000000000000484. Epub 2017 Mar 29. Erratum in: Circulation. 2019 Jul 30;140(5):e181-e184. PMID: 28356445.

Liu B, Hill HR. Role of Host Immune and Inflammatory Responses in COVID-19 Cases with Underlying Primary Immunodeficiency: A Review. J Interferon Cytokine Res. 2020 Dec;40(12):549-554. doi: 10.1089/jir.2020.0210. PMID: 33337932; PMCID: PMC7757688.

R: Thank you very much for your time involved in reviewing the manuscript and we also appreciate your clear and detailed feedback. We very much agree with your suggestions, and after carefully studying the literature you recommended, we have added parts to the introduction section and cited the references you recommended.

The corrections and additions are in the introduction section and are marked in red.

The additions are as follows: Kawasaki disease (KD) is an acute febrile disease of unknown etiology, characterized by systemic inflammation and vasculitis [1]. In contrast to Multisystem Inflammatory Syndrome in Children (MIS-C) related to Coronavirus disease 2019 (COVID-19), KD mostly occurs in children over five years of age. The most common complication of KD is coronary artery disease, especially coronary artery lesions (CALs) such as coronary artery dilation, aneurysm, coronary artery thrombosis and stenosis, life-threatening coronary aneurysm rupture, myocardial infarction, and heart failure in severe cases [2,3]. The use of intravenous immunoglobulin (IVIG) and aspirin in KD has significantly decreased the incidence of CALs, but CALs are still observed in 5–20% of patients in the acute phase [4–6]. Nevertheless, the pathogenesis of CALS due to KD is currently unclear, and no validated biomarkers are available to predict the occurrence of this complication. Therefore, it is challenging for clinicians to predict the occurrence of CALs and implement effective intervention measures. In this study, we aimed to explore the efficacy of clinical indicators combined with S100A12/TLR2-related signaling molecules in predicting KD with CALs and provide a basis for understanding the pathogenesis of CALs, and assist clinical decision-making.

A member of the calcium-binding S100 family, S100A12 activates inflammatory responses by binding to multiple receptors extracellularly in damage-related molecular patterns [7]. Previous studies have found that neutrophils in the early stage of KD can secrete S100A12 [8] to promote KD coronary artery lesions through the synergistic activation of endothelial cells [9,10], whereas S100A12 strictly depends on pattern recognition receptors, such as toll like receptors (TLRs), to function in vivo [11,12]. The S100A12 and TLR signaling molecules have not yet been reported as indicators of KD. TLRs are important in innate immunity, immunodeficiency, and COVID-19 [13,14]. The application of TLRs, cytokines, and other biomarkers in severe COVID and MIS-C (similar to KD) may provide a good example to illustrate the progress of novel biomarker application in rare or severe infections [15]. TLR2 has been reported to predict CAL progression in patients with KD [16,17], and a study by Soo et al. found that the high expression of TLR2 in single-nucleated cells was correlated with CALs and non-response in KD [16]. Highly expressed TLR can activate the signaling mediators domain-containing adapter-induced interferon-β (TRIF) and Myeloid differentiation primary response gene 88 (MYD88), allowing nuclear factor kappa-B (NF-kB) activation to promote TNF-α, IL-1, IL-6, and other pro-inflammatory factors [18]. Previous studies have confirmed that TLR2 and MyD88 contribute to Lactobacillus casei extract-induced focal coronary arteritis in a mouse model of KD [19], and Seyed et al. found that TLR2, TLR3, TLR9, MYD88, and TRIF gene transcript levels were upregulated in KD patients before IVIG treatment and downregulated after treatment [20]. In addition, several studies have found that NF-κB is involved in the development and progression of KD by participating in the inflammatory response, regulating the release of inflammatory factors [21], participating in immune activation [22], and inducing vascular endothelial damage [23]. Therefore, we envision that S100A12/TLR2 may activate the immune response of KD and may be involved in the development of CALs in KD by inducing the high expression of NF-κB by MYD88.

2. For inclusion criteria, what did the author mean by "complete data"?

 R: Thanks for pointing out the inaccurate description. "complete data" means “The case information is complete”, therefore ' complete data ' has been corrected to ‘case information’.

3. Is the statistical analysis two-tailed or one-tailed?

 R: Thanks for your comments! Although some clinical indicators have been proved to be valuable in the diagnosis of KD, its optimal cut-off point, diagnostic efficacy, and the value of the combined S100A12/TLR2 diagnostic model were still unclear, thus the two-tailed test statistical analysis was more favored chosen to be used here.

4. What reverse transcriptase (RT) did the authors use? What internal control gene did the RT-PCR of S100A12, TLR2, MYD88, and NFKB benchmark to? The differences in the expression of NF-κB and TLR2 between the two groups were not statistically significant. What are the limit of detection or analytical sensitivity of these RT-PCR? According to the following ref, different RTs with varied efficiency may cause unreliable results. Please discuss this based on this reference.

Liu B, Forman M, Valsamakis A. Optimization and evaluation of a novel real-time RT-PCR test for detection of parechovirus in cerebrospinal fluid. J Virol Methods. 2019 Oct;272:113690. doi: 10.1016/j.jviromet.2019.113690. Epub 2019 Jul 5. PMID: 31283959.

R: Thanks for your comments! According to your suggestions, we have carefully read the above literatures. In this study, the author compares the detection value of four commercial kits for parechoviruses (HPeV), including the performance characteristics of the detection system, influencing factors and other aspects, and finally concludes the optimal enzyme volume and the best approach to add MS2, which is a relatively comprehensive comparison of detection methods. However, there are many differences between this study and the paper. First, this study focuses on relatively stable genes (S100A12, TLR2, MYD88, and NF-κB), which are different from viruses with more genetic diversity, so it is not necessary to consider the detection status of different subtypes. Second, the detection of the virus needs to consider the detection limit, sensitivity, specificity of the reagent and various influencing factors in the experiment are closely related. In this study, the main detection targets are typical key molecules in the NF-κB/TLR2 pathway, and they also have certain expression levels in normal health’s, but they are significantly increased in diseases, so the detection of these genes is relatively less affected by detection factors in the same methodology (Exp Ther Med. 2022 Jun 30;24(3):543.; Front Immunol. 2022 Jan 19; 12:767512.; Avicenna J Med Biotechnol. 2022 Jul-Sep;14(3):188-195.). Third: this paper was to detect the expression of virus in cerebrospinal fluid, with poor sensitivity, while in this study (J Clin Microbiol. 2017 Jul;55(7):2035-2044.), we detected the expression of mononuclear cells in blood, with high sensitivity. We apologize for the lack of detail in the presentation of our materials and methods, and we have corrected it accordingly in methods and marked it in red.

RNA was extracted from the PBMC samples using TRIzol reagent and stored at -70℃, and the RNA concentration and purity were measured with the NanoDrop ND-1000 (Thermo Fisher Scientific Inc., Massachusetts, USA) spectrophotometer. RNAs were converted into cDNA by reverse transcription using the PrimeScript TM RT Reagent kit (Vazyme Biotech [Nanjing] Co., Ltd. CHINA). Following cDNA synthesis, real-time polymerase chain reaction (PCR) and relative quantification were performed using Premix Ex Taq SYBR (Vazyme Biotech [Nanjing] Co., Ltd.) to evaluate S100A12, TLR2, MYD88, and NF-κB gene expression. A SYBR Green Master Mix (20 µl; Invitrogen; Vazyme Biotech [Nanjing] Co., Ltd.) was used for real-time PCR amplification. The amplification conditions were as follows: 95°C for 30 s, followed by 40 cycles of 95°C for 3–10 s and 60°C for 10–30 s. The expression levels were calculated using the 2‑ΔΔCT method, and the levels were normalized against glyceraldehyde-3-phosphate dehydrogenase (GAPDH) as the control blank gene. Results are expressed as fold-changes. Gene-specific primer sequences were designed by Sangon Biotech Co. Ltd. (Shanghai, China) and are listed in Table 1. All experiments were performed at least three times.

---

## [Editor Report · Decision Letter 1]

27 Sep 2023

Clinical indicators combined with S100A12/TLR2 signaling molecules to establish a new scoring model for coronary artery lesions in Kawasaki disease

PONE-D-23-26094R1

Dear Dr. Ding,

We’re pleased to inform you that your manuscript has been judged scientifically suitable for publication and will be formally accepted for publication once it meets all outstanding technical requirements.

Kind regards,

Benjamin M. Liu, MBBS, PhD, D(ABMM), MB(ASCP)

Academic Editor

PLOS ONE
---

## [Editor Report · Acceptance letter]

3 Oct 2023

PONE-D-23-26094R1 

Clinical indicators combined with S100A12/TLR2 signaling molecules to establish a new scoring model for coronary artery lesions in Kawasaki disease 

Dear Dr. Ding:

I'm pleased to inform you that your manuscript has been deemed suitable for publication in PLOS ONE. Congratulations! Your manuscript is now with our production department. 

Kind regards, 

on behalf of

Dr. Benjamin M. Liu 

Academic Editor

PLOS ONE